# Integrative molecular characterization of sarcomatoid and rhabdoid renal cell carcinoma

Ziad Bakouny [1], David A. Braun [1], Sachet A. Shukla [2], Wenting Pan[1], Xin Gao[3], Yue Hou[2], Abdallah Flaifel[4], Stephen Tang [1], Alice Bosma-Moody[1], Meng Xiao He[1], Natalie Vokes [1], Jackson Nyman[1], Wanling Xie[5], Amin H. Nassar [1], Sarah Abou Alaiwi [1], Ronan Flippot[1], Gabrielle Bouchard[1], John A. Steinharter[1], Pier Vitale Nuzzo [1], Miriam Ficial [4], Miriam Sant'Angelo[4], Juliet Forman[1,2,6], Jacob E. Berchuck [1], Shaan Dudani[7], Kevin Bi[1], Jihye Park[1], Sabrina Camp[1], Maura Sticco-Ivins[4], Laure Hirsch[1], Sylvan C. Baca[1], Megan Wind-Rotolo[8], Petra Ross-Macdonald[8], Maxine Sun[1], Gwo-Shu Mary Lee[1], Steven L. Chang[1], Xiao X. Wei[1], Bradley A. McGregor[1], Lauren C. Harshman[1], Giannicola Genovese[9], Leigh Ellis [4,10], Mark Pomerantz[1], Michelle S. Hirsch[4], Matthew L. Freedman[1], Michael B. Atkins[11], Catherine J. Wu [1,6], Thai H. Ho [12], W. Marston Linehan [13], David F. McDermott [14], Daniel Y. C. Heng[7], Srinivas R. Viswanathan [1], Sabina Signoretti[4,10], Eliezer M. Van Allen [1,15✉] & Toni K. Choueiri [1,15✉]

Sarcomatoid and rhabdoid (S/R) renal cell carcinoma (RCC) are highly aggressive tumors with limited molecular and clinical characterization. Emerging evidence suggests immune checkpoint inhibitors (ICI) are particularly effective for these tumors, although the biological basis for this property is largely unknown. Here, we evaluate multiple clinical trial and real-world cohorts of S/R RCC to characterize their molecular features, clinical outcomes, and immunologic characteristics. We find that S/R RCC tumors harbor distinctive molecular features that may account for their aggressive behavior, including *BAP1* mutations, *CDKN2A* deletions, and increased expression of *MYC* transcriptional programs. We show that these tumors are highly responsive to ICI and that they exhibit an immune-inflamed phenotype characterized by immune activation, increased cytotoxic immune infiltration, upregulation of antigen presentation machinery genes, and PD-L1 expression. Our findings build on prior work and shed light on the molecular drivers of aggressivity and responsiveness to ICI of S/ R RCC.

[1] Department of Medical Oncology, Dana-Farber Cancer Institute, Boston, MA, USA. [2] Translational Immunogenomics Laboratory, Dana-Farber Cancer Institute, Boston, MA, USA. [3] Department of Medicine, Massachusetts General Hospital Cancer Center, Boston, MA, USA. [4] Department of Pathology, Brigham and Women's Hospital, Boston, MA, USA. [5] Department of Data Sciences, Dana-Farber Cancer Institute, Boston, MA, USA. [6] Broad Institute of MIT and Harvard, Cambridge, MA, USA. [7] Tom Baker Cancer Centre, University of Calgary, Calgary, AB, Canada. [8] Bristol-Myers Squibb, Princeton, NJ, USA. [9] Department of Genomic Medicine, The University of Texas MD Anderson Cancer Center, Houston, TX, USA. [10] Department of Oncologic Pathology, Dana-Farber Cancer Institute, Boston, MA, USA. [11] Lombardi Comprehensive Cancer Center, Georgetown University Medical Center, Washington, DC, USA. [12] Division of Hematology and Medical Oncology, Mayo Clinic, Scottsdale, AZ, USA. [13] Urologic Oncology Branch, Center for Cancer Research, National Cancer Institute, NIH, Bethesda, MD, USA. [14] Beth Israel Deaconess Medical Center, Boston, MA, USA. [15] These authors contributed equally: Eliezer M. Van Allen and Toni K. Choueiri. ✉email: eliezerm_vanallen@dfci.harvard.edu; Toni_Choueiri@dfci.harvard.edu

Sarcomatoid and rhabdoid (S/R) renal cell carcinoma (RCC) are among the most aggressive forms of kidney cancer[1,2]. Sarcomatoid and rhabdoid features represent forms of dedifferentiation of RCC tumors and can occur in the same tumor or independently of each other[3]. These features can develop over any background RCC histology, including clear cell, papillary, and chromophobe RCC. These tumors account for 10–15% of RCC and most patients with S/R RCC present with metastatic disease[1,4]. While classic RCC therapies such as VEGF and mTOR targeted therapies are largely ineffective for these tumors, multiple clinical studies suggest that immune checkpoint inhibitors (ICI) may have significant clinical activity in sarcomatoid and rhabdoid RCC[5–11]. Prior studies have hinted that these tumors may harbor distinctive molecular features, although these studies were limited by small sample sizes, restricted molecular analyses, leading to discordant conclusions[6,12–15].

To define the molecular properties underlying the S/R clinical subtype and determine their relationship to potentially enhanced response to ICI, we perform an expanded clinical and molecular integrated characterization of S/R RCC in both clinical trial and real-world cohorts, assessing clinical outcomes on ICI, genomic and RNA sequencing (RNA-seq), immunohistochemical (IHC) staining for PD-L1, immunofluorescence (IF)-based assessment of immune infiltration, and transcriptomic evaluation of sarcomatoid cell lines (Fig. 1a).

## Results

**S/R RCC tumors harbor distinctive genomic features**. We first evaluated the genomic landscape of S/R RCC (total $N = 208$) in three distinct cohorts (two whole exome sequencing [WES] and 1 gene panel sequencing cohort [OncoPanel]) and compared it to that of non-S/R RCC (total $N = 1565$; Supplementary Data 1). This DNA-sequencing cohort included one clinical trial WES cohort (CheckMate cohort; 69 S/R and 342 non-S/R), a retrospective analysis of an institutional panel-based sequencing cohort (OncoPanel cohort; 79 S/R and 395 non-S/R), and a retrospective pathologic review and analysis of a publicly available cohort (TCGA cohort; 60 S/R and 828 non-S/R). The most commonly altered genes in S/R RCC (Supplementary Fig. S1) were generally similar to those previously reported for RCC[16]. We subsequently compared the genomic features of S/R RCC tumors to background histology-matched non-S/R RCC tumors across the three cohorts. Tumor mutational burden (TMB), total indel load, and frameshift indel load were overall similar between S/R RCC and non-S/R RCC tumors (Supplementary Fig. S2a–c). While the frameshift indel load was significantly increased ($p = 0.024$) in S/R vs. non-S/R RCC in the OncoPanel cohort, the absolute difference was small (S/R vs. non-S/R means: 1.32 vs. 0.85 frameshift indels/Mb) and was not corroborated in the two WES cohorts (CheckMate and TCGA; Supplementary Fig. S2c).

Next, gene-specific alteration rates were compared between S/R and non-S/R RCC in each of the three cohorts independently and in combination (Methods). *BAP1* and *NF2* somatic alterations were significantly and consistently enriched in S/R compared to non-S/R RCC, whereas *KDM5C* somatic alterations were significantly less frequent in S/R compared to non-S/R RCC (Fisher's exact $q < 0.05$; Fig. 1b and Supplementary Data 2). Furthermore, *CDKN2A* and *CDKN2B* deep deletions as well as *EZH2* and *KMT2C* high amplifications were significantly enriched in S/R compared to non-S/R (Fisher's exact $q < 0.05$ and consistent across at least two of the three included datasets; Fig. 1b and Supplementary Data 2). Other genes that were significantly amplified (low or high amplification) included *MYC* and *CCNE1*, whereas those that were significantly deleted (shallow or deep deletion) included *RB1* and *NF2* (Fisher's exact

$q < 0.05$). Although recent reports have suggested that genes in the 9p24.1 locus (including *CD274*, *JAK2*, and *PCD1LG2* genes) were more frequently amplified in RCC tumors with sarcomatoid features[6,17], we did not observe focal amplifications to be enriched at this locus in these cohorts (Supplementary Data 2). Moreover, differences between S/R and non-S/R RCC were generally consistent regardless of background histology (clear cell or non-clear cell; Supplementary Data 2).

Since the analyses in this study are based on single region sampling of S/R RCC tumors and since such sampling has been shown to affect the detection rate of mutations in RCC tumors[18], we next compared the intra-tumoral heterogeneity (ITH) index between S/R and non-S/R RCC tumors (Methods). We found that the ITH index was not significantly different between these two groups of tumors in the CheckMate cohort ($p = 0.48$). Furthermore, this observation was corroborated in a re-analysis of the TRACERx Renal study, whereby the ITH index did not differ between S and non-S RCC tumors ($p = 0.21$; Supplementary Fig. S3a). Moreover, among 71 S/R RCC tumors in the OncoPanel cohort (of a total of 79 S/R RCC tumors) for which the portion of the tumor that was sequenced was assessable, 44 tumors had the S/R (mesenchymal) regions sequenced and 27 had the non-S/R (epithelioid) regions of the tumor sequenced. These two subsets of tumors were compared and no significant overall mutation/ indel load (Supplementary Fig. S3b) or gene-level mutational (Supplementary Data 3) differences were found, other than a marginal but statistically significant ($p = 0.042$) increase in the number of frameshift indels in mesenchymal regions. In addition, panel sequencing mutation data from 23 sarcomatoid tumors that had been laser micro-dissected (into sarcomatoid and epithelioid components) and sequenced separately from the study by Malouf et al.[19] was re-analyzed. In accordance, with the above findings no significant overall mutation/indel load (Supplementary Fig. S3c) or gene-level mutational (Supplementary Data 3) differences were found. However, it should be noted that alteration frequency for certain genes differed between mesenchymal and epithelioid portions of S/R RCC tumors (Supplementary Data 3). While certain mutations may be enriched in these tumors (in particular *TP53* mutations, as has been previously suggested[14]), none rose to the level of statistical significance in our cohort. Overall, our results suggest that the mutational differences between S/R and non-S/R RCC tumors are more pronounced than intra-tumoral mutational differences between mesenchymal and epithelioid portions of a given S/R RCC tumor. S/R RCC tumors have a distinctive genomic profile characterized by an enrichment for genomic alterations previously associated with poor prognosis in RCC (such as *BAP1* and *CDKN2A*) and genomic alterations that may represent therapeutic targets in S/R RCC (*CDKN2A* and *CDKN2B* deletions, *EZH2* amplifications, and *NF2* mutations).

**Transcriptomic programs of S/R RCC underpin their poor prognosis**. We next assessed transcriptomic programs in S/R RCC and their relationship to the known poor prognosis of this subtype. We compared RNA-seq data between S/R (total $N = 98$) and non-S/R RCC (total $N = 1077$) in the TCGA (publicly available; 59 S/R and 830 non-S/R) and CheckMate (39 S/R and 247 non-S/R) cohorts independently (Methods; Supplementary Data 4) using Gene Set Enrichment Analysis (GSEA)[20]. Twelve gene sets were upregulated (GSEA $q < 0.25$) in S/R compared to non-S/R RCC in the two cohorts independently, including cell cycle programs, genes regulated by *MYC*, and apoptosis programs (Fig. 2a; Supplementary Data 5). Specific upregulated gene sets may account for their morphological features including their mesenchymal appearance[3] (upregulation of epithelial-mesenchymal-transition [EMT]) and frequent co-occurrence of necrosis (endoplasmic

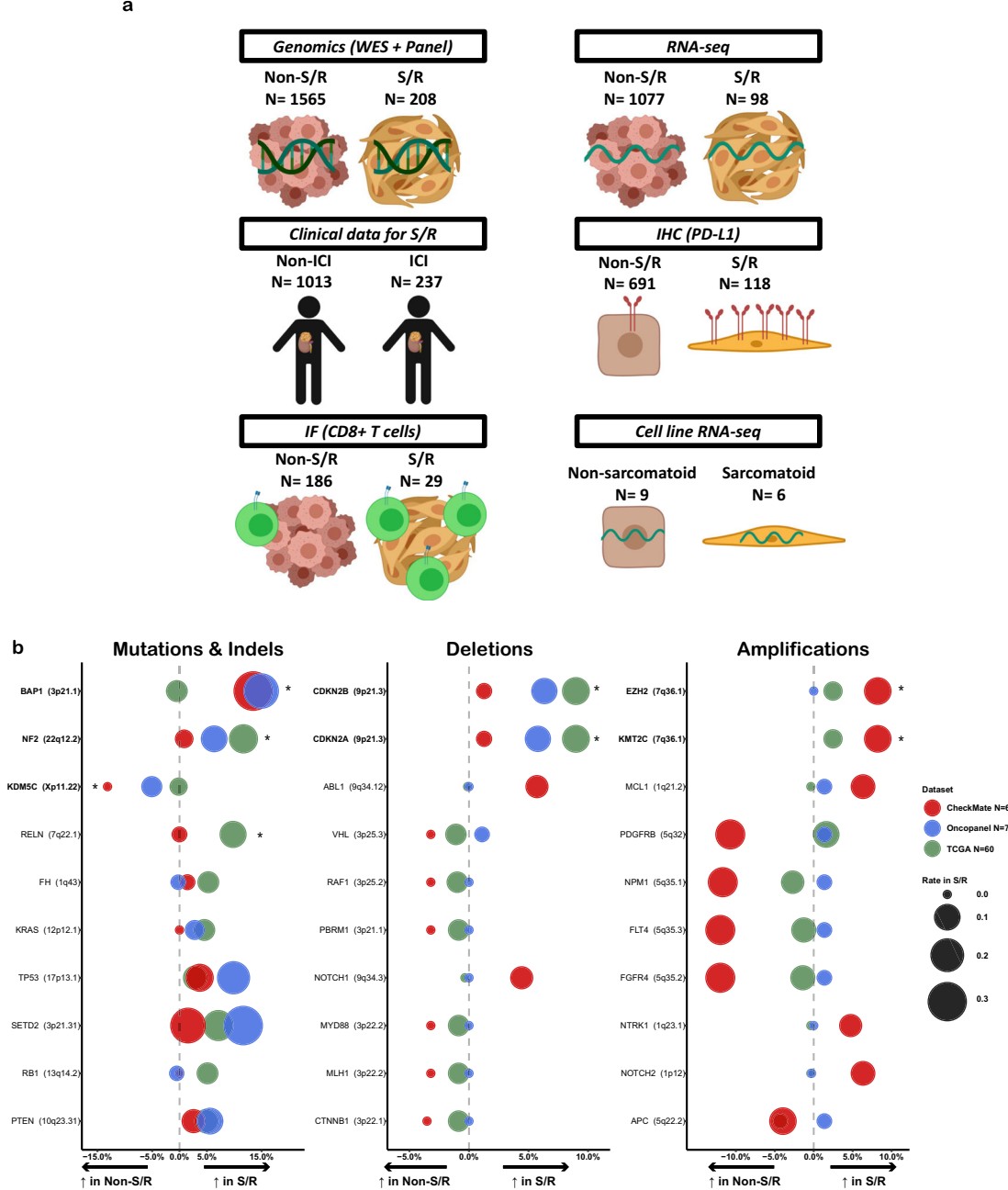

**Fig. 1 Genomic characterization of S/R RCC reveals distinctive genomic features. a** Overview of the clinical, molecular, and cell line data. (**b**) Comparison of S/R vs. non-S/R RCC by mutations & indels, deletions, and amplifications in the CheckMate, OncoPanel, and TCGA cohorts. *$q < 0.05$ (Fisher's method meta-analysis of Fisher's two-sided exact tests); ICI: Immune Checkpoint Inhibitor; IF: Immunofluorescence; IHC: Immunohistochemistry; RNA-seq: RNA-sequencing; S/R: Sarcomatoid/Rhabdoid; TCGA: The Cancer Genome Atlas; WES: Whole Exome Sequencing.

reticulum [ER] stress and apoptosis-caspase pathway)[1,4], and rapid progression (E2F targets, G2/M checkpoint, mitotic spindle assembly). Moreover, high *MYC* targets version 1 (v1) expression as quantified by single sample GSEA (ssGSEA) scores[21] significantly correlated with worse clinical outcomes in both the subset of patients with S/R in the anti-PD-1 (nivolumab) arm of the CheckMate cohort as well as the subgroup of stage IV S/R RCC patients in TCGA independently (Fig. 2b; Supplementary Fig. S4; Supplementary Data 6). Of note, the majority of founder gene sets of both the *MYC* v1 and v2 "Hallmark" gene were enriched in S/R RCC (Supplementary Fig. S5a), further corroborating the fact that *MYC*-regulated transcriptional programs are

enriched in S/R RCC. Moreover, the correlation with outcomes within S/R RCC of the *MYC* v1 score was consistent when the *MYC*-regulated transcriptional program was measured using the separate but related *MYC* v2 "Hallmark" gene set (Supplementary Fig. S5b, c). Patients with non-S/R RCC and *MYC* v1 scores similar to those of S/R RCC (above the median of the S/R RCC group for *MYC* v1) had significantly worse outcomes in both the TCGA and CheckMate PD-1 cohorts (Fig. 2c; Supplementary Fig. S4; Supplementary Data 6). These results indicate that a *MYC*-driven transcriptional program is driving the aggressive phenotype of S/R RCC tumors (also shared with a subset of non-S/R RCC)[2].

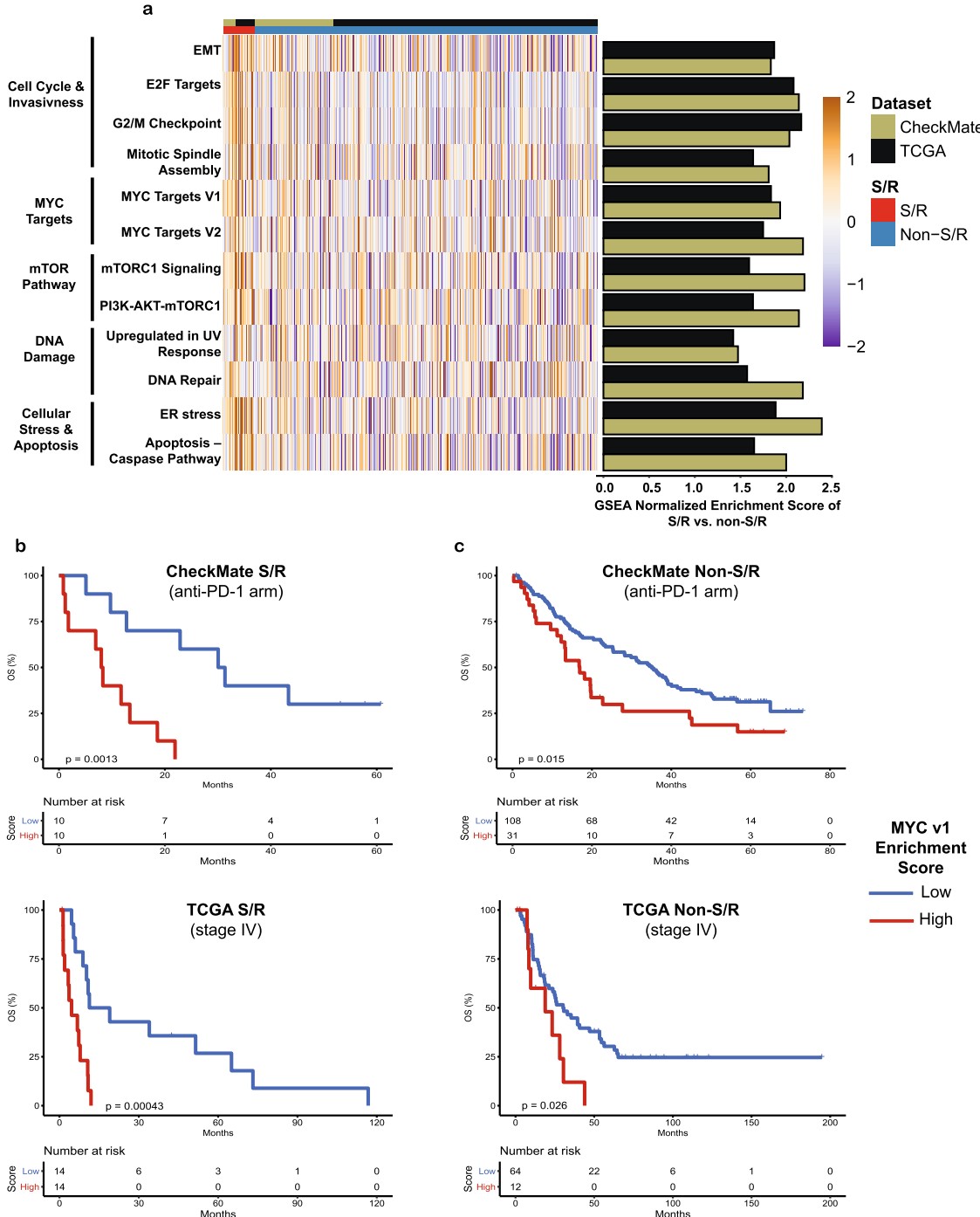

**Fig. 2 Transcriptional profiling of S/R RCC reveals the molecular correlates of its poor prognosis and identifies subsets of non-S/R tumors associated with a poor prognosis. a** Heatmap and bar plots of the ssGSEA scores and GSEA normalized enrichment scores for the non-immune "Hallmark" gene sets that were found to be significantly enriched ($q < 0.25$) in S/R compared to non-S/R RCC in both the TCGA and CheckMate cohorts independently. *P*-value calculated using a phenotype permutation-based two-sided test with 1000 permutations. Adjustments for multiple testing (50 "Hallmark" gene sets) were made using the false discovery rate (FDR) method. **b** Kaplan–Meier curves for OS by *MYC* v1 score within the S/R group of the CheckMate (anti-PD-1 arm) and TCGA (stage IV) cohorts; *MYC* v1 score dichotomized at the median. Log-rank test two-sided p-value reported without adjustment for multiple testing. **c** Kaplan–Meier curves for OS by *MYC* v1 score within the non-S/R group of the CheckMate (anti-PD-1 arm) and TCGA (stage IV) cohorts; *MYC* v1 score dichotomized at the median of the S/R group. Log-rank test two-sided p-value reported without adjustment for multiple testing. EMT: Epithelial Mesenchymal Transition; *MYC* v1: *MYC* Targets Version 1; S/R: Sarcomatoid/Rhabdoid; TCGA: The Cancer Genome Atlas.

Extending from the Hallmark GSEA analysis, 243 genes had significantly increased expression in S/R compared to non-S/R RCC independently across the two cohorts, including multiple cell cycle and proliferation (*CCNB1, CDC45, CDC6, CDCA3, CDCA7, CDCA8, CDK6,* and *MKI67*), immune (*HIVEP3, IFI16,*

*IFI35, IL15RA, and LAG3)*, and metastasis-implicated[22] (*ACTB, ANLN, ARPC1B, ARPC5,* and *ARPC5L, CD44)* genes as well as chemokine (*CXCL9*) and antigen presenting machinery (*TAP1, TAP2, CALR, PSMA5, PSMB10, PSMB4, PSMC2, PSME2*) genes that may be driving the immune infiltration in these tumors

(Supplementary Data 7). Since the overexpression of antigen presentation machinery genes has been found to correlate with increased cytotoxic immune infiltration and ICI responsiveness[23], we further explored the antigen presentation machinery genes using four dedicated REACTOME[24] and KEGG[25] gene sets and found all four to be significantly increased in both the CheckMate and TCGA cohorts independently (Supplementary Data 5). In addition, 83 genes had significantly decreased expression including cell junction-implicated (*TJP1* and *DSC2*) and cell differentiation genes (*MUC4*; Supplementary Data 7).

**S/R RCC tumors display an immune-inflamed phenotype.** With the unique molecular background of S/R RCC defined, we then sought to establish whether S/R RCC patients treated by immune checkpoint inhibitors (ICI) had improved clinical outcomes, as suggested by early studies, and whether particular molecular features established the basis for such clinical phenotypes. Patients with S/R RCC had improved outcomes on ICI compared to non-ICI agents across 3 cohorts (total N ICI arms = 237; total N non-ICI arms = 1013; Supplementary Data 8): a local Harvard cohort, the multicenter International Metastatic RCC Database Consortium (IMDC) cohort, and a pooled analysis of the S/R subgroup of 2 clinical trials (CheckMate 010[26] and CheckMate 025[27]) evaluating an anti-PD-1 agent (nivolumab) for metastatic RCC. Patients with S/R RCC had significantly improved outcomes on ICI compared to non-ICI across cohorts and clinical outcomes including overall survival (OS), progression free survival (PFS), time to treatment failure (TTF), and objective response rate (ORR; Fig. 3a–c).

Given the significant sensitivity of S/R RCC to ICI as reflected by improved responses and survival outcomes, we examined molecular features that may drive this phenotype. First, GSEA on the immune "Hallmark" gene sets of the RNA-seq data of the TCGA and CheckMate cohorts showed that all 8 "Hallmark" immune gene sets were enriched (GSEA $q < 0.25$) in S/R compared to non-S/R RCC in the two cohorts independently (Fig. 4a; Supplementary Data 4), including gene sets previously implicated in response to ICI (e.g. interferon gamma response)[28,29]. We then inferred immune cell fractions using the CIBERSORTx deconvolution algorithm (total N of S/R = 97 and Total N of non-S/R = 1028) and previously described gene signatures for Th1, Th2, and Th17 cells[30] in the RNA-seq data from the CheckMate and TCGA cohorts. CD8$^+$ T cell infiltration, CD8$^+$/CD4$^+$ T cell ratio, activated/resting NK cell ratio, M1 macrophages, M1/M2 macrophage ratio, as well as the Th1 score were all significantly increased (Mann–Whitney $q < 0.05$) in S/R RCC in both cohorts independently (Fig. 4b, Supplementary Fig. S6a; Supplementary Data 9). Moreover, the transcriptomic and immune microenvironment features of S/R RCC were consistent across S/R RCC subtypes (rhabdoid, sarcomatoid, or sarcomatoid and rhabdoid; Supplementary Figs. S7–9).

The immune-inflamed phenotype of S/R RCC tumors was further corroborated by an immunohistochemistry (IHC; N of S/R = 118 and N of non-S/R = 691) assay showing significantly increased PD-L1 (cut-off of ≥1%) expression on tumor cells in S/R compared to non-S/R tumors (43.2% vs. 21.0%; Fisher's exact $p < 0.001$; Fig. 4c and Supplementary Data 10) in the CheckMate cohort. To evaluate whether the elevated PD-L1 expression in S/R RCC is driven by PD-L1 gene amplification, as previously reported[6,17], we compared IHC-based PD-L1 expression by *CD274* (or PD-L1) gene copy number status (N = 63 patients in the S/R CheckMate cohort). We found that S/R tumors had increased PD-L1 expression (relatively to non-S/R RCC) independent of *CD274* copy number status (any deletion, amplification, or neither; all deletions were one-copy deletions); although the three S/R patients with *CD274* gene

amplification (1 patient with high amplification and 2 with low amplifications) all expressed PD-L1 by IHC above the cut-off of ≥1%. Moreover, *CD274* copy number status did not correlate with clinical outcomes in patients treated with a PD-1 inhibitor (Supplementary Fig. S10a–c). The immune-inflamed phenotype of S/R RCC tumors was also evaluated by IF staining for CD8$^+$ T cells in a subset of the CheckMate cohort (N of S/R = 29 and N of non-S/R = 186; Supplementary Fig. S6b, c and Supplementary Data 10). CD8$^+$ T cell infiltration at the tumor invasive margin, which had been reported to be associated with response to ICI-based therapies[31], tended to be increased in these tumors (although the difference was not statistically significant, Mann–Whitney $p = 0.14$). To further evaluate the immune microenvironment of S/R RCC tumors, we used immune/stroma signatures that are specific to RCC (empirical tumor microenvironment [eTME] signatures), as previously described[32]. In accordance with our other analyses, we found the eTME signatures tended to be enriched in S/R compared to non-S/R RCC (statistically significant for both the 20-fold and 3-fold signatures in the TCGA cohort and not significant in the CheckMate cohort; Supplementary Fig. S6d, e). Since *BAP1* mutations are enriched in S/R RCC tumors in this study and have been previously associated with immune infiltration and inflammation[32], we evaluated whether the immune findings reported in this study are only driven by *BAP1* mutations. In a sensitivity analysis excluding all *BAP1* mutants (from the S/R and non-S/R RCC) groups, the immune findings reported in this study were found to be largely consistent with the results of the primary analysis, suggesting that the immune findings of the current study in S/R RCC tumors are not solely driven by *BAP1* mutations (Supplementary Fig. S11). Taken together, S/R RCC tumors are highly responsive to ICI-based therapies and an immune-inflamed microenvironment in S/R RCC may be driving these responses in a manner that is not completely dependent on BAP1, leading to improved survival on ICI.

**Sarcomatoid cell lines recapitulate the biology of S/R RCC tumors.** To evaluate which transcriptomic programs enriched in S/R RCC tumors were attributable to sarcomatoid cancer cells rather than the microenvironment, we compared baseline RNA-seq data from 6 distinct sarcomatoid kidney cancer cell lines and 9 distinct non-sarcomatoid kidney cancer cell lines (Supplementary Fig. S12a, b; Supplementary Data 11). The transcriptional profile observed from the bulk profiling of tumors was partially recapitulated in the cell lines, with EMT and apoptosis-caspase pathway genes significantly enriched in sarcomatoid cell lines compared with non-sarcomatoid cell lines (Supplementary Fig. S12b). Given the shared transcriptional programs between sarcomatoid tumors and cell lines, we then sought to nominate candidate pathways that might reflect selective dependencies of sarcomatoid tumor cells. For this exploratory analysis, we interrogated publicly available data from 20 kidney cancer cell lines with both baseline RNA-seq and cell line drug response data. Among this group of 20 kidney cancer cell lines screened with 437 compounds of diverse mechanisms of action, we found EMT and apoptosis-caspase pathway ssGSEA scores most strongly correlated with sensitivity to cyclin dependent kinase inhibitors (CDKi; Supplementary Fig. S12c; Supplementary Data 11) and compared favorably to other classic therapeutic targets in RCC such as VEGF and mTOR inhibitors, consistent with the poor response of S/R RCC tumors to these agents[2,33]. In an attempt to corroborate these findings we focused on two CDKi agents, SNS-032 and alvocidib, that displayed a strong correlation of their sensitivity profiles with the EMT and apoptosis-caspase signature scores in CTRP (Supplementary Fig. S12; Supplementary

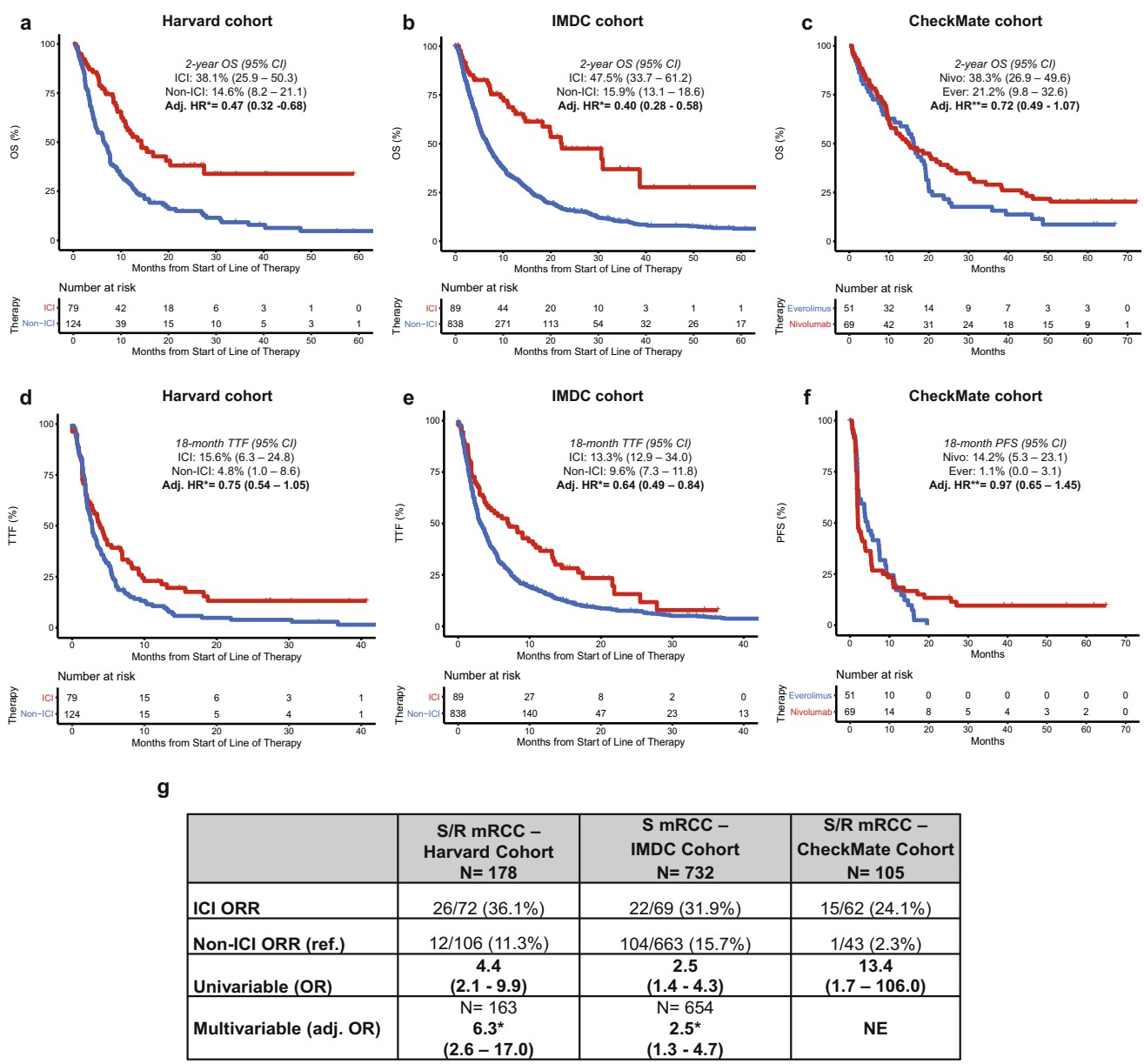

**Fig. 3 Improved clinical outcomes of S/R RCC tumors on immune checkpoint inhibitors across clinical trial and real-word cohorts.** OS on ICI compared to non-ICI in the **a** Harvard, **b** IMDC and **c** CheckMate S/R RCC cohorts. TTF on ICI compared to non-ICI in the **d** Harvard and **e** IMDC S/R RCC cohorts and **f** PFS in the CheckMate S/R RCC cohort. **g** Summary table of overall response rate (among evaluable patients) on ICI compared to non-ICI in patients with S/R RCC across the Harvard, IMDC, and CheckMate cohorts. 95% CI: 95% Confidence Interval; Adj. Adjusted; Ever: Everolimus; HR: Hazard Ratio; ICI: Immune Checkpoint Inhibitor; IMDC: International Metastatic Renal Cell Carcinoma Database Consortium; Nivo: Nivolumab; NE: Not Evaluable; OS: Overall Survival; S/R: Sarcomatoid/Rhabdoid. *Adjusted for IMDC (International Metastatic Renal Cell Carcinoma Database Consortium) risk groups, line of therapy, and background histology. **Adjusted for MSKCC (Memorial Sloan Kettering Cancer Center) risk groups.

Fig. S13a, b; Supplementary Data 11). In an independent in silico analysis of the recently published PRISM cell line drug screen dataset[34], a similar relationship between sensitivity to CDKi and the EMT and apoptosis signatures was found for alvocidib and other CDKi (Supplementary Fig. S13a; Supplementary Data 10; SNS-032 was not tested in the PRISM dataset). SNS-032, alvocidib, and a VEGF inhibitor control agent (axitinib) were also separately evaluated in two sarcomatoid RCC cell lines (UOK127 and RCJ41-T2; not included in the CTRP or PRISM screens) and three non-sarcomatoid RCC cell lines (Caki-2, KMRC-20, and KMRC-2; included in the CTRP or PRISM screens). Although the relative sensitivities for the non-sarcomatoid cell lines determined

in CTRP/PRISM globally mirrored relative sensitivities upon validation, we did not observe marked differential sensitivity between sarcomatoid and non-sarcomatoid cell lines for any of the 3 agents tested (Supplementary Fig. S14).

## Discussion

The current study represents a large integrative molecular and clinical characterization of S/R RCC, including clinical outcomes on ICI therapies and non-ICI controls from both clinical trial and retrospective cohorts, DNA and RNA-sequencing data, IHC and IF-based assessment of the immune microenvironment, and the

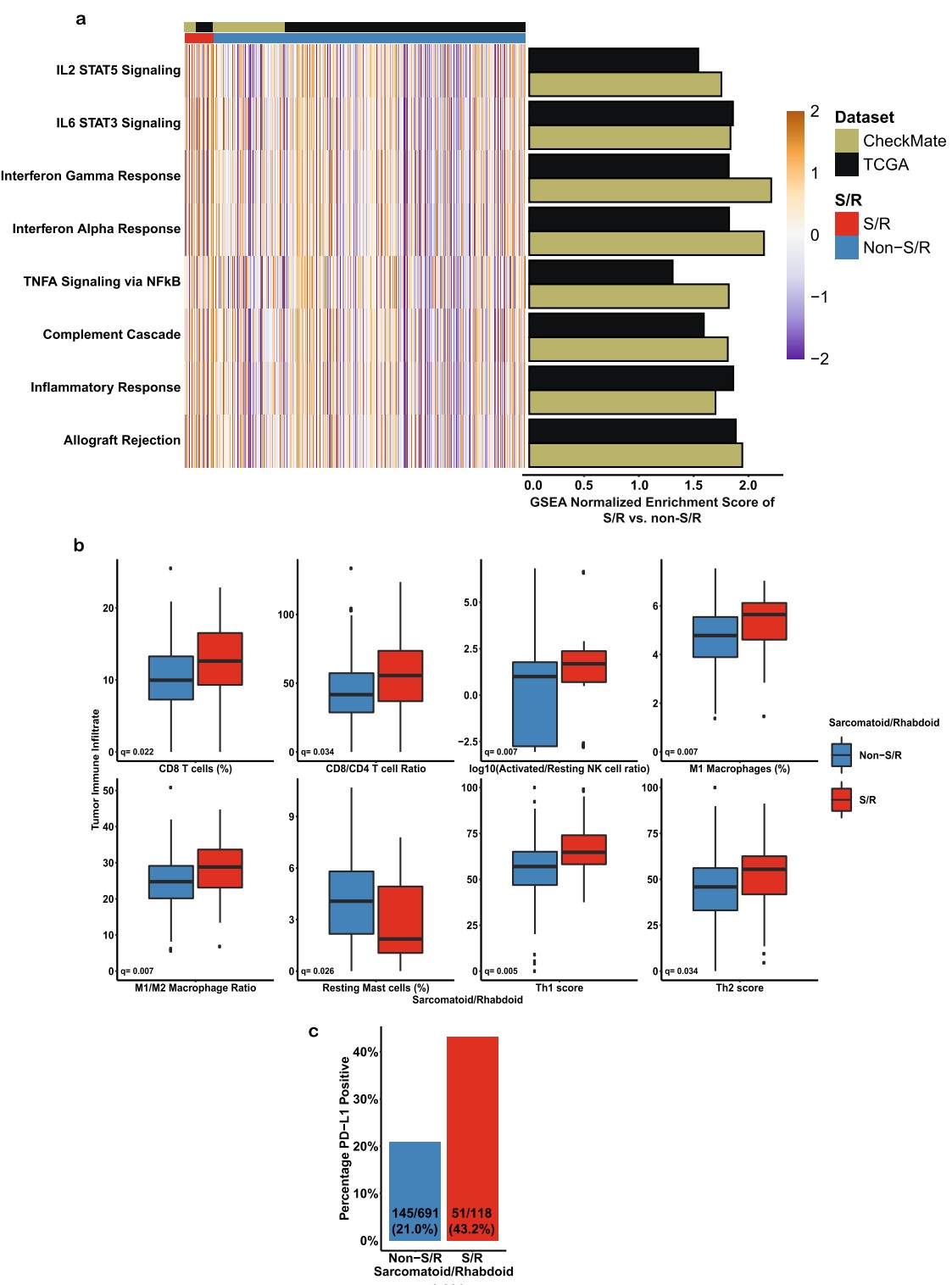

molecular profiling of cell line models of the disease. We show that S/R RCC tumors are highly responsive to ICIs, harbor distinctive genomic alterations, a characteristic transcriptional program characterized by the enrichment of *MYC*-regulated genes that correlates with poor outcomes, and a heavily inflamed microenvironment enriched in features that have been associated with ICI responses.

Our genomic findings corroborate those of prior studies that reported significant enrichment of Hippo pathway (which includes the *NF2* gene) mutations[19] in S vs non-S RCC tumors and *BAP1* mutations in S and R RCC tumors[12,15,35]. While *CDKN2A* alterations have been reported in S RCC tumors[13,19], these alterations are also present in non-S/R RCC tumors[36]. However, the current study established *CDKN2A/B* deep deletions as specifically enriched in S/R compared to non-S/R RCC tumors as well as depletion in *KDM5C* mutations and enrichment in *EZH2* amplifications in S/R RCC tumors. Moreover, S/R RCC tumors were not found to consistently

**Fig. 4 The immune-inflamed phenotype of S/R RCC tumors. a** Heatmap and bar plots of the ssGSEA scores and GSEA normalized enrichment scores for the immune "Hallmark" gene sets that were found to be significantly enriched (q < 0.25) in S/R compared to non-S/R RCC in the TCGA and CheckMate cohorts independently. *P*-value calculated using a phenotype permutation-based two-sided test with 1000 permutations. Adjustments for multiple testing (50 "Hallmark" gene sets) were made using the false discovery rate (FDR) method. **b** Boxplots of the comparison of CIBERSORTx and Th immune cell populations between S/R and non-S/R RCC, with two-sided Mann-Whitney U test comparisons corrected for multiple comparison testing (q value reported). Only variables which were significant (q < 0.05) in both the CheckMate and TCGA cohorts independently are shown. The CheckMate results are displayed in this figure (N = 39 S/R and N = 247 non-S/R RCC). The center of the box represents the median. The upper and lower hinges represent the 75th and 25th percentiles, respectively. The whiskers extend in both directions until the largest or lowest value not further than 1.5 times the interquartile range from the corresponding hinge. Outliers (beyond 1.5 times the interquartile range) are plotted individually. **c** Bar plot of the comparison of the proportions of tumors that were PD-L1 positive (≥1% on tumor cells) in S/R compared to non-S/R RCC. Two-sided Fisher's exact test p-value reported (*p* = 7.4×10⁻⁰⁷). TCGA: The Cancer Genome Atlas.

harbor a significantly increased rate of mutations, indels, or frameshift indels compared with non-S/R RCC tumors.

S/R RCC tumors are rapidly proliferating tumors that are associated with poor prognosis and rapid clinical progression[37,38]. While prior studies had identified multiple clinical and pathological factors that are associated with prognosis in patients with S/R RCC tumors[39,40], the molecular drivers of aggressivity of S/R RCC tumors had largely been unexplored. Here, we show that multiple molecular pathways implicated in cell cycle regulation and invasiveness as well as *MYC*-regulated genes are enriched in S/R RCC tumors and that the enrichment in *MYC*-regulated genes correlates with poor prognosis. These results suggest that *MYC*-regulated transcriptional programs are key factors driving the aggressivity and poor prognosis associated with S/R RCC tumors.

A previous study in genetically engineered mouse models had found that *MYC* activation with *CDKN2A* deletion and *VHL* deletion together produce kidney tumors that closely resemble human clear cell RCC[41]. While the authors of that study did not report histological patterns that resembled sarcomatoid and rhabdoid features, it is likely that these features are necessary but not sufficient to produce S/R features, and that other genomic and epigenomic features are needed to produce these aggressive tumors.

While prior studies have largely reported on tumors with sarcomatoid features, the different cohorts of this study highlight that rhabdoid features frequently co-occur with sarcomatoid features (10-20% of S/R RCC tumors). In addition, tumors harboring rhabdoid features alone are also relatively frequent (5-25% of S/R RCC tumors). In this study, the molecular features of S, R, and S + R (harboring both features concurrently) tumors were not found to be significantly different (Figure S1 and Figures S7-S9). However, detecting smaller effect sizes in these comparisons was limited by the relatively small sample sizes of the R and S + R groups.

The preliminary clinical outcomes of the subgroups of patients with S RCC from four large randomized clinical trials of the first line treatment of metastatic RCC[8–11] reported ORRs ranging between 46.8% and 58.8% for patients with S RCC treated with first line ICI combinations, with a significant clinical benefit compared to the non-ICI control arms (sunitinib in all four trials). These results for ICI arms are numerically superior to those reported in the current study (ORR range 24.1-36.1% in ICI arms). Multiple potential factors could account for the increased effectiveness observed in these subgroup analyses of phase III randomized controlled trials, compared to the findings in the three cohorts included in the current study. Indeed, the ICI arms in these studies were combination therapies (either PD-1 inhibitor + CTLA-4 inhibitor or PD-(L)1 + VEGF inhibitor) and all patients were being treated in the first line setting (and therefore not previously refractory to other therapies). In the current study, patients with S/R RCC derived significant clinical benefit from ICI regimens while having been treated by various different ICI regimens (entirely ICI monotherapy in the CheckMate cohort

and with a large proportion of ICI monotherapy in the IMDC and Harvard cohorts; Supplementary Data 7) and across different lines of therapy in each of the three cohorts (with a substantial proportion in the second line and beyond). Our findings, derived from three independent cohorts, suggest that S/R RCC tumors derive benefit from ICI regimens even outside of the setting evaluated in the subgroup analyses of the above-mentioned phase III trials (first line ICI combination regimens).

These recent data indicating that S RCC tumors are highly responsive to ICI have generated interest in determining the underpinnings of this responsiveness. Prior studies had suggested that S RCC tumors had increased tumor PD-L1 expression[42,43] and infiltration by CD8+ T cells[43]. These findings contrasted with another study that had reported that TGFβ signaling, which has been associated with immune exclusion and resistance to ICIs[44,45], was significantly increased in S RCC tumors[15]. More recently, two studies found that *CD274* (or PD-L1) gene amplifications are present in S RCC tumors and suggested that this genomic alteration may be underlying the increased PD-L1 tumor expression in these tumors and hypothesized that this genomic amplification may be underlying the immune responsiveness of S RCC tumors[6,17]. In the present study, the integrative analysis of WES, RNA-seq, tumor PD-L1 expression by IHC, tumor CD8+ T cell infiltration by IF, and clinical outcomes on ICI monotherapy from pre-treatment samples of patients with metastatic renal cell carcinoma on two clinical trials (CheckMate 010 and CheckMate 025) allowed the in-depth examination of the immune characteristics of these tumors. The present study corroborated the finding of increased PD-L1 tumor cell expression in S/R RCC and found that CD8+ T cell infiltration tended to be increased in these tumors. We did not find *CD274* gene focal amplification to be enriched in these tumors compared to non-S/R RCC tumors. The small number of S/R RCC tumors that harbored *CD274* gene amplification and had PD-L1 expression data available all expressed tumor cell PD-L1. However, the increased expression of tumor cell PD-L1 in S/R RCC tumors and the responsiveness of these tumors to PD-1 inhibitor monotherapy appeared to be independent of *CD274* gene amplification (Fig. 4c and Supplementary Fig. S10a–c). In addition, the analysis of two independent cohorts of RCC with RNA-seq (CheckMate and TCGA), revealed multiple previously unreported characteristics of the immune contexture of these tumors. First, all 8 "Hallmark" immune gene sets (but not the "Hallmark" TGFβ gene set), including IL6-JAK-STAT3 signaling and interferon gamma response, were enriched in S/R RCC tumors. Second, immune deconvolution revealed that multiple immune subsets that have previously been associated with an immune responsive microenvironment are significantly increased in S/R RCC tumors, including M1 macrophages, activated NK cells, and the Th1 T cell subset. These findings were also found to be largely consistent across S and R RCC subsets (Supplementary Figs. S8-9). Third, the expression of antigen presentation machinery genes, which

has been found to correlate with increased cytotoxic immune infiltration and ICI responsiveness[23], were significantly increased in S/R RCC tumors (Supplementary Data 5 and Supplementary Data 7).

In order to evaluate whether sarcomatoid cell line models recapitulate the biology of S/R RCC tumors, we compared the transcriptional profiles of 6 sarcomatoid cell lines to 9 non-sarcomatoid cell lines. Although less statistically powered to detect similar effect sizes to those observed in the bulk tumor S/R vs. non-S/R RCC comparison (due to a smaller sample size), the transcriptional programs of these cell lines partially recapitulated the biology of S/R RCC tumors. In particular, EMT and apoptosis-caspase pathway gene sets were significantly enriched in both S/R RCC tumors and sarcomatoid cell lines. These results suggest that at least some of the transcriptional findings reported in this study for S/R RCC are driven by the sarcomatoid tumor cells themselves and that sarcomatoid cell lines could serve as adequate models for these tumors in future therapeutic development efforts for this RCC subtype. Since the transcriptional programs of cell lines have been suggested to be most predictive of their sensitivity profiles (as opposed to other molecular features)[34,46], these two signatures were then projected into two independent cell line drug screen datasets (CTRP and PRISM)[34,47]. Sensitivity to CDK inhibitors appeared to correlate strongly with EMT and apoptosis-caspase pathway signatures in both datasets independently (Supplementary Fig. S12-13 and Supplementary Data 11). The CDK inhibitors that scored in these analyses target multiple CDKs, including those involved in transcription and cell cycle progression. We tested two CDKi (SNS-032 and alvocidib) along with a tyrosine kinase inhibitor control (axitinib) in two sarcomatoid and three non-sarcomatoid cell lines. The two sarcomatoid cell lines displayed decreased sensitivity to axitinib (a VEGF pathway inhibitor) as compared with the non-sarcomatoid cell line with the lowest EMT ssGSEA score, KMRC-20 (Supplementary Fig. S12b and S14c), underscoring the limited response to this inhibitor of this canonical clear cell RCC pathway[48] in these sarcomatoid cell lines. Sarcomatoid and non-sarcomatoid RCC cell lines showed globally similar sensitivities to the two CDKis tested in our assay. The overall sensitivity of both sarcomatoid and non-sarcomatoid RCC lines to the two CDKis tested may be explained by the specificities of the particular drugs tested as well as the plasticity in EMT gene expression program, even among non-S/R RCCs, that may modulate sensitivity to this class of agents. Study of the precise molecular determinants of response to these and other classes of therapeutic agents in S/R RCC is a ripe area for future investigation.

A limitation of this study is the potential bias induced by the inherent heterogeneity of S/R RCC tumors. Foci of sarcomatoid and rhabdoid features can be present anywhere within RCC tumors. When these tumors are being evaluated by pathologists, these foci of S/R features can be missed and S/R RCC tumors could be mis-classified as non-S/R RCC. In this study, we reviewed the pathology reports and slides of tumors (Methods) to attempt to minimize such misclassifications. Moreover, any biases due to misclassification would be expected to decrease the power of this study to detect an effect, thereby potentially increasing the risk of false negative but not false positive findings. In addition to misclassification, intra-tumoral histological heterogeneity (sarcomatoid/rhabdoid vs epithelioid foci within the same S/R RCC tumor in a patient) could also be associated with intra-tumoral molecular heterogeneity. In this study, using data from the present study and previously published studies, we find that the intra-tumoral mutational heterogeneity of S/R RCC tumors seems to be largely similar to that of non-S/R RCC tumors. In accordance with prior studies[14], we find that mutations in certain genes (in particular TP53) may be enriched in S/R components of

S/R RCC tumors. However, our overall analysis results suggest that mutational differences between S/R and non-S/R RCC tumors are greater than intra-tumoral mutational differences within S/R RCC tumors. The drivers of intra-tumoral histological heterogeneity require further evaluation and could be further investigated using novel single cell (DNA and/or RNA) and spatial transcriptomic methods. We additionally acknowledge that the data used in this study originated from cohorts that differed in the types of samples used (such as frozen tissue vs. formalin-fixed paraffin-embedded) and sequencing platform (panel vs. WES). Despite this heterogeneity, the differences between S/R and non-S/R RCC tumors were found to be consistent across the different cohorts.

In conclusion, our findings suggest that sarcomatoid and rhabdoid renal cell carcinoma tumors have distinctive genomic and transcriptomic features that may account for their aggressive clinical behavior. We also established that these tumors have significantly improved clinical outcomes on immune checkpoint inhibitors, which may be accounted for by an immune-inflamed phenotype; itself driven in part by upregulation of antigen presentation machinery genes in S/R RCC. Finally, our results suggest that sarcomatoid cell lines recapitulate the transcriptional programs of S/R RCC tumors and could serve as reasonably faithful models for these tumors, fueling the engine for future therapeutic discovery in this aggressive subtype of RCC. Further work is needed to determine whether other solid tumors with similar histological dedifferentiation components exhibit comparable molecular and clinical characteristics.

## Methods

**Clinical cohorts and patient samples**. The comparative clinical outcomes on immune checkpoint inhibitors (ICI) of patients with metastatic sarcomatoid and rhabdoid (S/R) renal cell carcinoma (RCC) were derived from: (1) CheckMate cohort (S/R RCC $N = 120$): two clinical trials evaluating an anti-PD-1 inhibitor (nivolumab) for metastatic clear cell RCC, CheckMate-025[27] (NCT01668784) and CheckMate-010[26] (NCT01354431), (2) Harvard cohort (S/R RCC $N = 203$): a retrospective cohort from the Dana-Farber/Harvard Cancer Center including patients from Dana-Farber Cancer Institute, Beth Israel Deaconess Medical Center, and Massachusetts General Hospital, (3) IMDC cohort (S/R RCC $N = 927$): a retrospective multi-center cohort of metastatic RCC that includes more than 40 international cancer centers and more than 10,000 patients with metastatic RCC. All patients had consented to an institutional review board (IRB) approved protocol to participate in the respective clinical trials and to have their samples collected for tumor and germline sequencing (for the CheckMate cohort) or to have their clinical data retrospectively collected for research purposes (Harvard and IMDC cohorts). Analysis was performed under a secondary use protocol, approved by the Dana-Farber Cancer Institute IRB. For all cohorts, the definition of sarcomatoid and rhabdoid RCC tumors was based on the ISUP 2013 consensus definitions: tumors were classified as harboring sarcomatoid features if they had any percentage of sarcomatoid component and as harboring rhabdoid features if they had any percentage of rhabdoid component (regardless of the background histology)[49]. For the Harvard and IMDC cohorts, sarcomatoid and rhabdoid status were determined by retrospective reviews of pathology reports. For the CheckMate cohort, sarcomatoid and rhabdoid features were retrospectively identified by review of pathology reports and of pathology slides by a pathologist. For the TCGA cohort, all pathology reports were first reviewed. Candidate sarcomatoid and/or rhabdoid cases were then reviewed by a pathologist. Cases that were unequivocal by the ISUP 2013 consensus definitions by pathology report and/or slide review were included. The TCGA cohort also included a subset of sarcomatoid RCC patients that had been previously retrospectively identified[15]. All pathology slides and reports for TCGA were accessed using cbioportal (https://www.cbioportal.org). Specifically, the following datasets were used: Kidney Renal Clear Cell Renal Cell Carcinoma (TCGA, Provisional), Kidney Chromophobe (TCGA, Provisional), Kidney Renal Papillary Cell Carcinoma (TCGA, Provisional). The sarcomatoid and rhabdoid annotations for the samples identified in TCGA are reported in Supplementary Data 12. The clinical characteristics of the patients in the CheckMate cohort with molecular sequencing data were similar to those of the overall trials[50].

**Cell Lines**. Fifteen cell lines were acquired by our laboratory for baseline RNA-seq characterization including 6 that had been derived from sarcomatoid kidney cancer tumors (RCJ41M, RCJ41T1, RCJ41T2, BFTC-909, UOK127, and UOK276) and 9 that had been derived from non-sarcomatoid kidney cancer tumors (786-O, A498, ACHN, Caki-1, Caki-2, KMRC-1, KMRC-2, KMRC-20, and VMRC-RCZ).

UOK127 and UOK276 were obtained from Dr. Linehan's laboratory at the National Cancer Institute (NCI) while RCJ41M, RCJ41T1, and RCJ41T2 were obtained from Dr. Ho's laboratory (Mayo Clinic, Phoenix, Arizona)[51]. Caki-1, Caki-2, A498, ACHN and 786-O were acquired from the American Type Culture Collection (ATCC). KMRC-1, KMRC-2, KMRC-20, VMRC-RCZ were obtained from JCRBbCell Bank and Sekisui XenoTech, LLC. BFTC-909 was obtained from Leibniz-Institut (DSMZ-Deutsche Sammlung von, Mikroorganismen und Zellkulturen GmbH).

Cell lines ACHN, VMRC-RCZ and 786-O were maintained in RPMI 1640 media (Gibco), supplemented with 10% FBS (Gibco) and 1% penicillin-streptomycin. Cell line A498 was maintained in EMEM media (Gibco), supplemented with 10% FBS (Gibco) and 1% penicillin-streptomycin. Caki-1 and Caki-2 were maintained in McCoy's 5 A media (Gibco), supplemented with 10% FBS (Gibco) and 1% penicillin-streptomycin. KMRC-1, KMRC-2, KMRC-20, UOK127, UOK276, BFTC-909, RCJ41T1, RCJ41T2 and RCJ41M were maintained in DMEM media (Gibco), supplemented with 10% FBS (Gibco) and 1% penicillin-streptomycin. Cultures were grown in a 37 °C incubator with 5% CO2. Total RNAs were isolated using the Trizol® reagent (Invitrogen), according to the manufacturer's instructions.

For cell viability assays, cells were seeded in 96-well plates at densities ranging from 1,000-10,000 cells per well, depending on the cell line. After 24 h, axitinib (S1005, Selleck), alvocidib (S1230, Selleck), or SNS-032 (S1145, Selleck) were added to cells at the indicated final concentrations. DMSO treatment was used as a negative control. Cell viability for 4 biological replicates of each treatment condition was assessed after 72 h after drug treatment using the CellTiter-Glo Luminescent Cell Viability Assay (G7571, Promega) and an EnVision Multilabel Plate Reader (PerkinElmer). Viability was calculated for each cell line relative to its respective DMSO control wells.

**RNA and DNA extraction, sequencing and pre-processing**. The methods used for DNA and RNA extraction and sequencing in the CheckMate 010 and 025 trials are described in a separate paper in more detail[50]. Briefly, archived formalin-fixed paraffin embedded (FFPE) tissue from pre-treatment samples of patients enrolled in these two trials were used. DNA and RNA were extracted from tumor samples along with paired germline DNA from whole blood. Germline and tumor DNA were sequenced using Illumina HiSeq2500 following a 2×100 paired-end sequencing recipe and targeting a depth of coverage of 100x. RNA was sequenced using a stranded protocol using Illumina HiSeq2500 following a 2×50 paired-end sequencing recipe and targeting a depth of 50 million reads. Mean exome-wide coverage for tumor samples was 129x and 112x for matched germline. For the RNA-seq data, the mean mapping rate of the included samples was 96.7% and mean number of genes detected was 21078.

For the TCGA cohort, publicly available data was downloaded for mutation data (https://gdc.cancer.gov/about-data/publications/mc3-2017), CNA data (https://www.cbioportal.org/datasets), upper-quartile (UQ) normalized transcripts-per-million (TPM) RNA-seq data (https://www.cbioportal.org/datasets), and clinical data (https://www.cbioportal.org/datasets)[52,53]. The dataset from the study by Malouf et al.[19] of paired sequencing of sarcomatoid RCC was downloaded from https://www.nature.com/articles/s41598-020-57534-5#Sec16 (supplementary dataset 1). The dataset from the TRACERx Renal study[18] was downloaded from https://www.ncbi.nlm.nih.gov/pmc/articles/PMC5938372/ (Supplementary Data 1 and Supplementary Data 2).

For the OncoPanel cohort, DNA extraction and sequencing were performed as previously described for the OncoPanel gene panel assay[54]. The OncoPanel assay is an institutional analytic platform that is certified for clinical use and patient reporting under the Clinical Laboratory Improvement Amendments (CLIA) Act. The panel includes 275 to 447 cancer genes (versions 1 to 3 of the panel), including 239 genes that are common across all 3 versions of the panel. Mean sample-level coverage for the Oncopanel cohort was 305x.

For the 15 cell lines acquired by our laboratory, RNA-seq was done using Illumina Platform PE150 polyadenylated non-stranded sequencing. The average mapping rate was 98.9% and 17998 genes were detected on average (all RNA-seqQC2 quality control metrics are reported in Supplementary Data 11). RNA-seq data (which were UQ normalized to an upper quartile of 1000 and log2-transformed) for 20 kidney cancer cell lines with RNA-seq and drug sensitivity data were downloaded from The Cancer Dependency Map Portal (DepMap)[55] (https://depmap.org/portal/download/) and drug sensitivity data were downloaded from the Cancer Therapeutics Response Portal (CTRP v2)[47] (https://portals.broadinstitute.org/ctrp/?cluster=true&page=#ctd2Cluster) and the PRISM 19Q4 secondary screen (https://depmap.org/portal/download/) as areas under the curve (AUC) for all agents.

**Genomic analysis**. The analytical pipeline for the WES data for the CheckMate 010 and 025 trials is described in detail in a separate paper[50]. Briefly, paired-end Illumina reads were aligned to the hg19 human genome reference using the Picard pipeline (https://software.broadinstitute.org/gatk/documentation/tooldocs/4.0.1.0/picard_fingerprint_CrosscheckFingerprints.php). Cross-sample contamination were assessed with the ContEst tool[56], and samples with ≥5% contamination were excluded. Point mutations and indels were identified using MuTect[57] and Strelka[58], respectively. Possible artifacts due to orientation bias, germline variants,

sequencing and poor mapping were filtered using a variety of tools including Orientation Bias Filter[59], MAFPoNFilter[60], and RealignmentFilter. Copy number events were called and filtered using GATK4 ModelSegments[61]. Copy number panel-of-normals was created based on matched germline samples. GISTIC[62] was used to determine gene-level copy number alteration events. Clonality assessment was performed using ABSOLUTE[63]. Mutations were considered clonal if the expected cancer cell fraction (CCF) of the mutation as estimated by ABSOLUTE was 1, or if the estimated probability of the mutation being clonal was greater than 0.5. The intratumor heterogeneity index (ITH) was defined as the ratio of subclonal mutations to clonal mutations.

OncoPanel mutation and gene-level copy number calling was performed as previously described[54]. In particular, variants were filtered to exclude those that occurred at a frequency of >0.1% in the Exome Sequencing Project database (http://evs.gs.washington.edu/EVS/) in order to remove variants that were probably germline variants. Additionally, in order to further remove potential germline variants from the OncoPanel results, Ensembl Variant Effect Predictor (VEP)[64] was run on the OncoPanel mutations and mutations present at an allelic frequency of 0.5% in one of the superpopulations from the 1000 Genomes Project[65] (https://www.internationalgenome.org/data) were excluded from all downstream analyses.

For the purposes of the present genomic analysis, mutation and CNA of 244 genes were analyzed (Supplementary Data 13), including the 239 genes that are common across the 3 versions of the panel, 3 frequently mutated genes in RCC (KDM5C, KMT2D, and PBRM1)[16] that are only included in versions 2 and 3 of the panel, and 2 genes that are included in none of the 3 versions of the panel, including a frequently mutated RCC gene (KMT2C)[16] and a gene that has been previously suggested to be more frequently mutated in sarcomatoid RCC (RELN)[15]. All mutations from TCGA, Oncopanel, and CheckMate cohorts were annotated using Oncotator[66] (https://software.broadinstitute.org/cancer/cga/oncotator). For WES data, only mutations with more than 30x coverage were included.

Somatic genomic alterations (mutations and insertions-deletions [indels]) were considered to be pathogenic if they were truncating (nonsense or splice site), indels, or missense mutations that were predicted to be pathogenic by Polyphen-2 HumDiv score[67] ≥0.957 or Mutation Assessor[68] score >1.90. Tumor mutational burden was calculated as the sum of all non-synonymous mutations divided by the estimated bait set (30 Megabases [Mb] for WES, 1.32 Mb for panel v3, 0.83 Mb for panel v2, and 0.76 Mb for panel v1). Moreover, the indel burden (either all indels or only frameshift indels) was normalized by dividing the indel by the estimated bait set for each version of OncoPanel. Gene-level deep deletions and high amplifications were considered for the primary copy number analysis, while any deletions (one-copy or two-copy) and any amplifications (low or high) were analyzed as a supplementary analysis.

The co-mutation plot was generated excluding patients that had either mutation or CNA data missing in any of the 3 cohorts (as reported in Supplementary Data 1). The estimate of percentage mutated took into account the missing genes for patients sequenced by panel sequencing (these percentages were estimated while excluding patients sequenced by panel sequencing for RELN and KMT2C, while only the patients sequenced by panel v1 were excluded for KDM5C, KMT2D, and PBRM1). TMB was compared between S/R and non-S/R in each of the three cohorts independently using Mann-Whitney U tests. Genomic alterations (mutations and indels, deep deletions, and high amplifications analyzed separately) were compared between S/R and non-S/R in each of the three cohorts independently using a Fisher's exact test. For the OncoPanel cohort, for KDM5C, KMT2D, and PBRM1, patients that had been sequenced by panel version 1 were excluded from the analysis. Only genes that were altered in at least 5% of patients (in all patients with RCC or in the S/R RCC group) in at least one of the 3 cohorts were tested. The p-values from the 3 cohorts were subsequently combined using Fisher's method for meta-analyses. The combined p-values were corrected for multiple hypothesis testing using Benjamini-Hochberg correction. Findings were considered to be significant if they were statistically significant at q < 0.05 and the same direction of the effect was observed in at least two of the three included datasets.

For the analysis of paired data in the dataset by Malouf et al. (paired sarcomatoid and epithelioid regions of S RCC tumors), continuous variables were compared by the paired Wilcoxon signed rank test. Mutation rates in genes were compared using McNemar's test.

**Transcriptomic analysis**. RNA-seq data from the CheckMate cohorts and the 15 cell lines sequenced in our laboratory were aligned using STAR[69], quantified using RSEM[70], and evaluated for quality using RNA-seqQC2[71]. Samples were excluded if they had an interquartile range of log2(TPM + 1)<0.5 or had less than 15,000 genes detected. Additionally, since the CheckMate cohort had been sequenced by a stranded protocol, samples were filtered if they had an End 2 Sense Rate<0.90 or End 1 Sense Rate>0.10 (as defined by RNA-seqQC2). For samples where RNA-seq was performed in duplicates, the run with a higher interquartile range of log2(TPM + 1), considered a surrogate for better quality data, was used. We subsequently filtered genes that were not expressed in any of the samples (in each cohort independently) then UQ-normalized the TPMs to an upper quartile of 1000, and log2-transformed them. Since the CheckMate cohort had been sequenced in 4 separate batches, principal component analysis (PCA) was used to evaluate for batch effects and 4 batches were observed. These 4

batches were corrected for using ComBat[72] (Supplementary Fig. S15). Subsequently, a PCA was performed on the ComBat-corrected expression matrix to confirm that batch effects had been adequately corrected for (Supplementary Fig. S15). Moreover, a constant that was equal to the first integer above the minimum negative expression value obtained post-ComBat (constant of +21) was added to eliminate negative gene expression values that were a by-product of ComBat correction. The ComBat-corrected expression matrix was used for all downstream analyses on the CheckMate cohort. All downstream analyses were computed on the TCGA and CheckMate cohorts independently and only results which were found to be independently statistically significant in each of the two cohorts were considered to be significant.

GSEA between S/R and non-S/R was run using the Java Application for GSEA v4.0.0 and MSigDB 7.0[73] on the 50 "Hallmark" gene sets, MYC v1 and v2 "Founder" gene sets, and select KEGG[25] and REACTOME[24] antigen presentation machinery gene sets. Gene sets were considered to be enriched if $q < 0.25$. Single sample GSEA (ssGSEA) was additionally computed using the "GSVA" package[21] in the R programming environment to obtain sample-level GSEA scores. Differential gene expression analysis was computed using the non-parametric Mann-Whitney U test and Benjamini-Hochberg false discovery rate correction with $q < 0.05$ considered statistically significant. The CIBERSORTx deconvolution algorithm[74] was used to infer immune cell infiltration from RNA-seq data (Job type: "Impute cell fractions"), in absolute mode, on the LM22 signature[75], with B mode batch correction (in order to correct for the batch effect between the LM22 signature, which was derived from microarray data, and the data used in this study which consisted of RNA-seq), with quantile normalization disabled, and in 1000 permutations. All samples which had a p-value for deconvolution >0.05 were considered to have failed deconvolution and were therefore discarded from all downstream analyses. Relative cell proportions were obtained by normalizing the CIBERSORTx output to the sample-level sum of cell counts (in order to obtain percentages of immune infiltration). A constant of $10^{-06}$ was added to all proportions in order to allow the computation of immune cell ratios. Additionally, Th1, Th2, and Th17 scores were computed using ssGSEA (and were normalized to scores between 0 and 100) based on previously described signatures for these cell types[30]. All immune cell proportions and ratios were compared between S/R and non-S/R using a non-parametric Mann-Whitney U test with Benjamini-Hochberg correction and a q-value threshold of 0.05 for statistical significance.

In order to evaluate whether specific signatures predicted outcomes in S/R RCC, Cox regression models were performed to evaluate the relationship between ssGSEA scores, modeled as continuous variables (multiplied by a factor of 100), and survival outcomes. ssGSEA scores found to be significantly associated with survival outcomes were used to dichotomize S/R RCC patients into two groups at the median of the score. The dichotomized groups were evaluated using Kaplan-Meier curves and compared using log-rank tests. In order to evaluate whether such relationships held in patients with non-S/R RCC, the same analysis was conducted in non-S/R RCC using the ssGSEA scores that were found to be related to outcomes in S/R RCC. In addition, for non-S/R RCC patients, the group was also dichotomized based on the median of the S/R RCC group and compared by Kaplan-Meier methodology and log-rank tests. In particular, this was done for MYC v1 scores which were found to be significantly related to outcomes in the S/R RCC group and not found to be related to outcomes when evaluated continuously in the non-S/R RCC group or when dichotomized at the median.

**Cell line in silico drug sensitivity analysis.** In order to evaluate potential novel therapeutic targets for S/R RCC, we computed ssGSEA scores for the 20 kidney cancer cell lines in DepMap that also had drug sensitivity data reported as areas under the curve (AUCs) of the dose-response curve in CTRP v2 and in the PRISM secondary screen. Using the gene signatures that were found to be significantly upregulated in both bulk tumor RNA-seq cohorts (in the TCGA and CheckMate cohorts independently) and sarcomatoid cell lines, we correlated the scores to drug sensitivity AUC data using Pearson's r correlation coefficients. Only therapeutic agents that were tested in at least 8 of the 20 kidney cancer cell lines were evaluated in CTRP v2. For visualization, the ssGSEA-AUC correlations were grouped by drug types and illustrated in a heatmap (in which negative correlations indicated that higher ssGSEA scores correlated with lower AUCs and therefore greater sensitivity). Moreover, scatter plots of the correlations were displayed for key correlations.

**Immunohistochemistry and immunofluorescence.** PD-L1 expression on the membrane of tumor cells was assessed using the Dako assay, as previously described in the CheckMate 025 and 010 trials[26,27]. Tumors were considered PD-L1 positive if they expressed PD-L1 on ≥1% of tumor cells.

The immunofluorescence assay used is described in detail in a separate paper[50]. CD8 immunostain was performed as part of a multiplex fluorescent IHC panel on 4 µm FFPE sections. Tumor sections were stained using the Opal multiplex IHC system (PerkinElmer), which is based on tyramide-conjugated fluorophores. All slides were counterstained with Spectral DAPI (PerkinElmer) and manually coverslipped. The slides were imaged using the Vectra 3 automated quantitative

pathology imaging system (PerkinElmer) and whole slide multispectral images were acquired at 10x magnification.

Digital whole slide multispectral images were then uploaded into HALO Image Analysis platform version 2.1.1637.18 (Indica Labs). For each case, the tumor margin and center were defined while also excluding empty spaces, necrosis, red blood cells and fibrotic septa. Specifically, the tumor margin was defined as the space within 500 µm (in either direction) of the interface between the tumor and surrounding tissue. Image analysis algorithms were built using Indica Labs High-Plex FL v2.0 module to measure the area within each layer, perform DAPI-based nuclear segmentation and detect CD8 (FITC)-positive cells by setting a dye cytoplasm positive threshold. A unique algorithm was created for each tumor and its accuracy was validated through visual inspection by at least one pathologist.

**Clinical outcomes.** For patients in the Harvard and IMDC cohorts, clinical data were retrospectively collected. OS was defined as the time from the start of the line of therapy (ICI or non-ICI) until death from any cause. Time to treatment failure (TTF) was defined as the time from start of the line of therapy until discontinuation of therapy for any cause. Since assessment of responses in these retrospective cohorts was not subject to radiological review specifically for the purpose of this study, responses were defined based on RECIST v1.1 criteria[76] as available by retrospective review. For the CheckMate cohort, OS was defined from the time of randomization until death from any cause. Progression free survival (PFS) was defined from randomization until death or progression. Both PFS and ORR were defined using RECIST v1.1 criteria. All patients who were lost to follow-up or did not have an event at last follow-up were censored.

**Statistical analysis.** The dose-response curves for the in vitro cell viability assays performed at DFCI were generated using GraphPad PRISM 8. All analyses were done in the R programming environment 3.6.1. For boxplots, the upper and lower hinges represent the 75th and 25th percentiles, respectively. The whiskers extend in both directions until the largest or lowest value not further than 1.5 times the interquartile range from the corresponding hinge. Outliers (beyond 1.5 times the interquartile range) are plotted individually. Continuous variables were summarized by their means and standard deviations (SD) or medians and interquartile ranges (IQR) or ranges. Categorical variables (such as gene alterations) were summarized by their percentages. For survival outcomes, the Kaplan-Meier methodology was used to summarize survival distributions in different groups; 18-month PFS (or TTF) and 2-year OS were provided with 95% confidence intervals. For survival outcomes, multivariable Cox regression models were used for the comparison of ICI and non-ICI regimens and adjusted hazard ratios (HR) with their 95% confidence intervals were reported. Specifically, the IMDC risk groups[77] (Poor vs. Intermediate/Favorable), line of therapy (2nd line and beyond vs. 1st line), and background histology (clear cell vs. non-clear cell) were adjusted for in the Harvard and IMDC cohort analyses and the Memorial Sloan Kettering Cancer Center (MSKCC) risk groups[78] (Poor vs. Intermediate vs. Favorable) were adjusted for in the CheckMate cohort analysis. Similarly, the ORR was compared between the ICI and non-ICI using multivariable logistic regression models adjusting for the same covariates (except for the CheckMate cohort, in which only one patient had had a response in the everolimus arm and therefore the adjusted odds ratio was not estimable). For all multivariable analyses, patients with missing data in any of the variables were excluded from the analysis. For ORR analyses, only patients who were evaluable for response were included in the analysis. The Kaplan–Meier methodology for assessing point estimates of survival was computed using the "landest" package in R. All heatmaps were created using the R package "pheatmap" and were computed using Z-score transformations. When multiple cohorts were represented in the same heatmap, the Z score normalization was done within each cohort separately (in order to account for batch effects in visualization). All tests were two-tailed and considered statistically significant for $p < 0.05$ or $q < 0.05$ unless otherwise specified.

**Reporting summary.** Further information on research design is available in the Nature Research Reporting Summary linked to this article.

## Data availability

All relevant data are available from the authors and/or are included with the manuscript. All clinical and correlative data from the CheckMate 010 and 025 clinical trials are made separately available as part of the accompanying paper[50]. WES data from the CheckMate 010 and 025 clinical trials from patients who consented to deposition have been submitted to the European Genome-phenome Archive (Accession numbers EGAS00001004291 and EGAS00001004292). All intermediate data from the RNA-seq analyses of the CheckMate and TCGA cohorts are made available in Supplementary Data 6 (single sample gene set enrichment analysis scores) and Supplementary Data 9 (CIBERSORTx immune deconvolution). The raw, transformed, and intermediate data from the generated cell line RNA-seq data are made available in Supplementary Data 11. The clinical data from the Harvard cohort are available in Supplementary Data 14. For the TCGA cohort, publicly available data was downloaded for mutation data (https://gdc.cancer.gov/about-data/publications/mc3-2017), CNA data (https://www.cbioportal.org/datasets), RNA-seq data (https://www.cbioportal.org/datasets), and clinical data (https://

www.cbioportal.org/datasets). The dataset from the study by Malouf et al. of paired sequencing of sarcomatoid RCC was downloaded from https://www.nature.com/articles/s41598-020-57534-5#Sec16 (supplementary dataset 1). The dataset from the TRACERx Renal study was downloaded from https://www.ncbi.nlm.nih.gov/pmc/articles/PMC5938372/ (Supplementary Data 1 and Supplementary Data 2).RNA-seq data for 20 kidney cancer cell lines with RNA-seq and drug sensitivity data were downloaded from The Cancer Dependency Map Portal (DepMap) (https://depmap.org/portal/download/) and drug sensitivity data were downloaded from the Cancer Therapeutics Response Portal (CTRP v2) (https://portals.broadinstitute.org/ctrp/?cluster=true?page=#ctd2Cluster) and the PRISM 19Q4 secondary screen (https://depmap.org/portal/download/) as areas under the curve (AUC) for all agents. Exome Sequencing Project database (http://evs.gs.washington.edu/EVS/) and 1000 Genomes Project data (https://www.internationalgenome.org/data) were used to detect potential germline variants from tumor-only gene panel sequencing data. MSigDB 7.0 (https://www.gsea-msigdb.org/gsea/msigdb) was used to define gene pathways of interest. Any other queries about the data used in this study should be directed to the corresponding authors of this study.

## Code availability

Algorithms used for data analysis are all publicly available from the indicated references in the Methods section. Any other queries about the custom code used in this study should be directed to the corresponding authors of this study.

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

## Acknowledgements

We thank the OncoPanel study team and the patients who contributed their data to research and participated in clinical trials. We thank all contributors to The International Metastatic Renal-Cell Carcinoma Database Consortium for their data contributions. We thank Dr. Kanishka Sircar and Dr. Jose Antonio Karam for providing us with additional data and insights from their previously published study. This work was supported in part by Dana-Farber / Harvard Cancer Center Kidney Cancer SPORE (P50-CA101942-12), DOD CDMRP (W81XWH-18-1-0480), and Bristol-Myers Squibb. S.A.S acknowledges support by the NCI (R50RCA211482). R.F. has been funded in part by the ARC foundation grant. X.X.W. has been funded in part by the KCA YIA. C.J.W. is supported in part by The G. Harold and Leila Y. Mathers Foundation, and she is a Scholar of the Leukemia and Lymphoma Society. T.H.H. is supported by National Cancer Institute (R01 CA224917). S.R.V. was supported by the Claudia Adams Barr Program for Innovative Cancer Research. E.M.V is supported by NIH R01 CA227388. T.K.C. is supported in part by the Dana-Farber/Harvard Cancer Center Kidney Cancer SPORE and Program, the Kohlberg Chair at Harvard Medical School and the Trust Family, Michael Brigham, and Loker Pinard Funds for Kidney Cancer Research at DFCI.

## Author contributions

Conception and Design: Z.B., D.A.B., E.M.V.A., T.K.C. Provision of study material or patients: Z.B., W.P., G.-S.M.L., S.T., K.B., J.P., S.C., M.S.-I., L.H., S.C.B., G.G., M.S.H., S.R.V., S.L.C., X.X.W., B.A.M., L.C.H., L.E., M.P., M.L.F., M.B.A., C.J.W., T.H.H., W.M.L., D.F.M., D.Y.C.H., S.S., E.M.V.A. and T.K.C. Collection and assembly of data: Z.B., S.T., S.R.V., X.G., A.F., A.H.N., S.A.A., P.R.-M., S.C.B., R.F., G.B., J.A.S., P.V.N., M.F., M.S.'A., J.E.B., S.D. Data Analysis and Interpretation: Z.B., S.T., S.R.V., D.A.B., S.A.S., Y.H., W.X., P.R.-M., Alice B.-M., M.X.H., N.V., J.N., J.F., M.S., E.M.V.A., T.K.C. Manuscript writing and revision: All authors. Final approval of manuscript: All authors. Accountable for all aspects of work: All authors

## Competing interests

Z.B.: reported nonfinancial support from Bristol-Myers Squibb and Genentech/ImCore. D.A.B. reported nonfinancial support from, Bristol-Myers Squibb, and personal fees from Octane Global, Defined Health, Dedham Group, Adept Field Solutions, Slingshot Insights, Blueprint Partnerships, Charles River Associates, Trinity Group, and Insight Strategy, outside of the submitted work. S.A.S. reported nonfinancial support from Bristol-Myers Squibb, and equity in 152 Therapeutics outside the submitted work. X.G: Research Support to Institution: Exelixis. X.X.W: Research Support: BMS. B.A.M is a consultant for Bayer, Astellas, Astra Zeneca, Seattle Genetics, Exelixis, Nektar, Pfizer, Janssen, Genentech and EMD Serono. He received research support to Dana Farber Cancer Institute (DFCI) from Bristol Myers Squibb, Calithera, Exelixis, Seattle Genetics. L.C.H reports consulting fees from Genentech, Dendreon, Pfizer, Medivation/Astellas, Exelixis, Bayer, Kew Group, Corvus, Merck, Novartis, Michael J Hennessy Associates (Healthcare Communications Company and several brands such as OncLive and PER), Jounce, EMD Serrano, and Ology Medical Education; Research funding from Bayer, Sotio, Bristol-Myers Squib, Merck, Takeda, Dendreon/Valient, Jannsen, Medivation/Astellas, Genentech, Pfizer, Endocyte (Novartis), and support for research travel from Bayer and Genentech. M.B.A: Advisory Board participation: BMS, Merck, Novartis, Arrowhead, Pfizer, Galactone, Werewolf, Fathom, Pneuma, Leads BioPharma; Consultant: BMS, Merck, Novartis, Pfizer, Genentech-Roche, Exelixis, Eisai, Aveo, Array, AstraZeneca, Ideera, Aduro, ImmunoCore, Boehringer-Ingelheim, Iovance, Newlink, Pharma, Surface, Alexion, Acceleron, Cota, Amgen; Research Support to institution: BMS, Merck, Pfizer, Genentech. C.J.W.: Co-founder of Neon Therapeutics, and is a member of its SAB. Receives research funding from Pharmacyclics. T.H.H: Advisory board participation: Surface Therapeutics, Exelixis, Genentech, Pfizer, Ipsen, Cardinal Health; research support-Novartis. D.F.M reports a consulting/advisory role for Bristol-Myers Squibb, Merck, Roche/Genentech, Pfizer, Exelixis, Novartis, Eisai, X4 Pharmaceuticals, and Array BioPharma; and reports that his home institution receives research funding from Prometheus Laboratories. D.H: consulting or research funding from Pfizer, Novartis, BMS, Ipsen, Exelixis, and Merck. S.R.V.: consultant for MPM Capital and has consulted for Vida Ventures. S.S: Research support to Institution: Bristol-Myers Squibb, AstraZeneca, Novartis, Exelixis; Consultant: Merck, AstraZeneca, Bristol-Myers Squibb, AACR, and NCI; royalties: Biogenex. E.M.V: Advisory/Consulting: Tango Therapeutics, Genome Medical, Invitae, Illumina, Ervaxx; Research support: Novartis, BMS; Equity: Tango Therapeutics, Genome Medical, Syapse, Ervaxx, Microsoft; Travel reimbursement: Roche/Genentech; Patents: Institutional patents filed on ERCC2 mutations and chemotherapy response, chromatin mutations and immunotherapy response, and methods for clinical interpretation. T.K.C.: Research (Institutional and personal): Alexion, Analysis Group, AstraZeneca, Bayer, Bristol Myers-Squibb/ER Squibb and sons LLC, Calithera, Cerulean, Corvus, Eisai, Exelixis, F. Hoffmann-La Roche, Foundation Medicine Inc., Genentech, GlaxoSmithKline, Ipsen, Lilly, Merck, Novartis, Peloton, Pfizer, Prometheus Labs, Roche, Roche Products Limited, Sanofi/Aventis, Takeda, Tracon. Honoraria: Alexion, American Society of Medical Oncology, Analysis Group, AstraZeneca, Bayer, Bristol Myers-Squibb/ER Squibb and sons LLC, Cerulean, Clinical Care Options, Corvus, Eisai, EMD Serono, ESMO, Exelixis, F. Hoffmann-La Roche, Foundation Medicine Inc., Genentech, GlaxoSmithKline, Harborside Press, Heron Therapeutics, Inc (Healthcare Communications Company with several brands such as OnClive, PeerView and PER), Jansen Oncology, IQVIA, Ipsen, Kidney Cancer Journal, Lancet Oncology, Lilly Oncology, L-path, Medscape, Merck, Michael J. Hennessy (MJH) Associates, Navinata Healthcare, NCCN, NEJM, Novartis, Peloton, Pfizer, Platform Q, Practice, PrimeOncology, Prometheus Labs, Research to, Roche, Roche Products Limited, Sanofi/Aventis, Up-to-Date. Consulting or Advisory Role: Alexion, Analysis Group, AstraZeneca, Bayer, Bristol Myers-Squibb/ER Squibb and

sons LLC, Cerulean, Corvus, Eisai, EMD Serono, Exelixis, Foundation Medicine Inc., Genentech, GlaxoSmithKline, Heron Therapeutics, Infinity Pharma, Ipsen, Lilly, Lilly Ventures, Merck, NCCN, Novartis, Peloton, Pfizer, Pionyr, Prometheus Labs, Roche, Sanofi/Aventis, Surface Oncology, Tempest, Up-to-Date. Stock ownership: Pionyr, Tempest. No leadership or employment in for-profit companies. Other present or past leadership roles: Director of GU Oncology Division at Dana-Farber and past president of Medical Staff at Dana-Farber), member of NCCN Kidney panel and the GU Steering Committee, past chairman of the Kidney Cancer Association Medical and Scientific Steering Committee), KidneyCan Advisory board, Kidney Cancer Research Summit co-chair (2019-), various ASCO/ESMO roles on educational and scientific committees. Patents filed, royalties or other intellectual properties: related to biomarkers of immune checkpoint blockers, and circulating free methylated DNA Travel, accommodations, expenses, in relation to consulting, advisory roles, or honoraria. Medical writing and editorial assistance support may have been funded by Communications companies funded by pharmaceutical companies (e.g. ClinicalThinking, Envision Pharma Group, Fishawack Group of Companies, Health Interactions, Parexel, Oxford PharmaGenesis, pharmagenesis, and others). The institution (Dana-Farber Cancer Institute) may have received additional independent funding of drug companies or/and royalties potentially involved in research around the subject matter. CV provided upon request for scope of clinical practice and research. Mentored several non-US citizens on research projects with potential funding (in part) from non-US sources/Foreign Components. No speaker's bureau.
