## [Peer Review File · Nature Communications]

Reviewers' Comments:

Reviewer #1:

Remarks to the Author:

Bakouny et al., performed a pooled study of diverse RCC patient cohorts (real world and clinical trial) with variable genomic and other information to investigate sarcomatoid/rhabdoid biology and to assess response to ICIs.

While the study is limited by the diversity of the cohorts and associated approaches, as well as in the amount of new data, the scope of the study and, in key instances, reproducibility of the results are assets that may justify publication in *Nat Commun*.

Major points:

Justification for the grouping of sarcomatoid and rhabdoid tumors should be provided.

The authors conduct a sensitivity analysis and conclude that the inflamed immune environment and response to ICI is independent of BAP1 loss. However, sensitivity analyses are suboptimal to address this question, and are better poised to address for the presence of other contributing factors. Furthermore, Fig 4b and Fig S11 are not comparable. In addition, while FDR-adjusted q values are reported in Fig 4b, p values are reported for Fig S11, suggesting that analyses in Fig S11 are not corrected for multiple comparisons. These analyses should be revised to adjust for multiple comparisons. In addition, the authors should directly address the question they posed about the association of BAP1 loss with an inflamed immune environment and response to ICI.

Related to the study they cite showing a link between BAP1 loss and inflammation, it would be preferable to adapt the empirically defined TME signature to evaluate the contribution of the stroma rather than the use of sarcomatoid and non-sarcomatoid cancer cell lines.

The added value of the drug studies is unclear since no differential response between sarcomatoid and non-sarcomatoid lines was found.

Authors should clarify the data and expand the discussion about differences across the cohorts they analyze. For example, in Fig S2 there are differences across the cohorts, but this is not so obvious given the different units and small axes font sizes.

Reviewer #3:

Remarks to the Author:

Van Allen, Choueiri and coworkers report the molecular characterization of sarcomatoid and rhabdoid RCC in a comprehensive manner by comprehensive DNA and RNA sequencing across an impressive number of tumors, some of which come from a clinical trial. While there have been past reports molecularly characterizing S/R RCC none have performed it to this degree and in such a large cohort and in such cutting edge manner. The authors find that specific alterations are enriched in S/R RCC. For example, CDKN2A deletion, and MYC gene signature upregulation. Moreover, tumors are enriched in an inflamed phenotype with high PD-L1 expression and increased clinical benefit from ICI. The latter point holds great clinical import, is based on multiple datasets, including data from the CheckMate trial, and will likely be practice changing (despite being retrospective).

Specific Points.

Pg 5, Line 102. When describing the non-S/R RCC cohorts it would be good to indicate the numbers of tumors in each of the outlined cohorts.

Multiple genomic features of S/R RCC found by the authors have interestingly been explored in a prior GEM model of RCC [i.e. VHL loss, CDKN2A deletion, MYC overexpression] (PMID: 28593993). It is interesting that the authors of that paper did not see a S/R RCC phenotype in the mouse model. It seems worth a couple sentences (perhaps within or at the end of the middle paragraph on page 14) in the discussion mentioning this and the likelihood that other genetic or epigenetic events (i.e. Tp53 mutation) are required for a full S/R phenotype.

Questions and Responses

Referee #1 (Remarks to the Author):

Summary	Bakouny et al., performed a pooled study of diverse RCC patient cohorts (real world and clinical trial) with variable genomic and other information to investigate sarcomatoid/rhabdoid biology and to assess response to ICIs. While the study is limited by the diversity of the cohorts and associated approaches, as well as in the amount of new data, the scope of the study and, in key instances, reproducibility of the results are assets that may justify publication in Nat Communic.
Q1	Justification for the grouping of sarcomatoid and rhabdoid tumors should be provided.
Response	We thank the reviewer for the opportunity to clarify this point. There are four major reasons why sarcomatoid and rhabdoid tumors were grouped:  - Both forms of tumor represent de-differentiation in RCC (PMID: 24025520). - Both forms of tumor have been found to be associated with adverse prognosis in RCC (PMID: 24025520, 25450036, 32070319). - The two forms very often co-occur with up to 1/3 of sarcomatoid tumors harboring a rhabdoid component and up to 1/2 of rhabdoid tumors harboring a sarcomatoid component (as evidenced in analyses of our own cohorts; Table S1). When the two forms do co-occur, the two components are often inter-mixed together pathologically (PMID: 12460207). - In analyses of molecular data of this study, no apparent differences were found between sarcomatoid alone, rhabdoid alone, and sarcomatoid + rhabdoid tumors (Figures S1, S7, S8, and S9). As we highlight in the discussion, this does not rule out that molecular differences may exist but that these differences appear to be subtle. Overall, sarcomatoid and rhabdoid features were grouped in this study because they share similar clinical, pathological, and molecular characteristics, in addition to the fact that they are often inter-mixed when they do co-occur.
Q2	The authors conduct a sensitivity analysis and conclude that the inflamed immune environment and response to ICI is independent of BAP1 loss. However, sensitivity analyses are suboptimal to address this question, and are better poised to address for the presence of other contributing factors.
Response	The authors thank the reviewer for the opportunity to clarify this point.

	The authors do not wish to suggest that BAP1 loss does not correlate with an inflamed microenvironment. We agree with the reviewer that this has been previously established in RCC in general (such as in PMID: 29884728). Rather, our sensitivity analyses (in response to the reviewer’s question during the first round of revisions) aimed to show that the immune-inflamed phenotype of S/R RCC (compared to non-S/R RCC) is not solely driven by BAP1 loss. We believe that our sensitivity analyses, in which we excluded all patients that had BAP1 mutant tumors from both the S/R and non-S/R RCC groups and re-conducted the same analysis, are appropriate to assess this specific question. The concordance of the results of our original analyses including all patients (Figures 4 and S6) and our sensitivity analyses excluding all BAP1 mutants from both the S/R and non-S/R RCC groups (Figure S11) demonstrate that the immune-inflamed phenotype of S/R RCC is not solely driven by BAP1 mutations. However, in order to clarify the purpose of our analysis we have removed the term “in a BAP1-independent manner” from the Results section on page 12, line 274. The section in question now reads: “Since BAP1 mutations are enriched in S/R RCC tumors in this study and have been previously associated with immune infiltration and inflammation³², we evaluated whether the immune findings reported in this study are only driven by BAP1 mutations. In a sensitivity analysis excluding all BAP1 mutants (from the S/R and non-S/R RCC) groups, the immune findings reported in this study were found to be largely consistent with the results of the primary analysis, suggesting that the immune findings of the current study in S/R RCC tumors are not solely driven by BAP1 mutations (Fig. S11). Taken together, S/R RCC tumors are highly responsive to ICI-based therapies and an immune-inflamed microenvironment in S/R RCC may be driving these responses in a manner that is not completely dependent on BAP1, leading to improved survival on ICI.”
Q3	Furthermore, Fig 4b and Fig S11 are not comparable. In addition, while FDR-adjusted q values are reported in Fig 4b, p values are reported for Fig S11, suggesting that analyses in Fig S11 are not corrected for multiple comparisons. These analyses should be revised to adjust for multiple comparisons.
Response	We thank the reviewer for this comment. As highlighted in the response to Q2 above, we do not claim that Figure 4b (or Figure S6) and Figure S11 are entirely interchangeable. However, we aim to show in Figure S11, that the immune inflamed phenotype of S/R RCC is not entirely driven by BAP1 mutants, as re-emphasized in the text and discussed above. We had reported p-values since this analysis is not testing all variables but specifically testing whether the 8 comparisons that were found to be

significant (after q-value correction) in two independent cohorts are *solely* driven by *BAP1* mutants.

Importantly, independently of any statistical testing, we find that the same trends that had been observed in the overall cohort were conserved in each of the two cohorts after removing all *BAP1* mutants from both the S/R and non-S/R RCC groups. Statistical testing in Figure S11 serves only to further reinforce this point by showing that, despite the decreased statistical power to detect the same effect size that inevitably occurs when removing patients from both cohorts (the sensitivity analysis removes 19% of patients from the CheckMate cohort and 28% of patients from the TCGA cohort), we find many of the comparisons to be statistically significant in the two cohorts after removing all *BAP1* mutants (and all samples for which *BAP1* mutation status could not be determined due to the lack of matching WES).

While this does not negate prior studies (such as PMID: 29884728) that have established the relationship between *BAP1* mutations and an immune inflamed phenotype in RCC in general, it does show that within S/R RCC specifically, the immune inflamed phenotype that we observed is not solely driven by *BAP1* mutations.

However, for consistency and in accordance with the reviewer's comment, we have now **updated Figure S11** to report q-values adjusting for the comparisons performed in the sensitivity analyses in each cohort. The results of the updated analysis are largely concordant with our original sensitivity analysis reporting p-values. The updated figure panels are the following two figure panels:

Q4 In addition, the authors should directly address the question they posed about the association of *BAP1* loss with an inflamed immune environment and response to ICI.

Response We thank the reviewer for the suggestion. In order to evaluate directly the relationship between *BAP1* loss and the immune-inflamed phenotype of S/R RCC (the 8 immune cell populations reported above), we directly compared within S/R RCC, tumors with *BAP1* loss-of-function (LOF) mutations to tumors without *BAP1* LOF mutations, we found to significant differences between the two groups. Moreover, no consistent trend was found towards an immune-inflamed phenotype in tumors with *BAP1* LOF mutations compared to tumors without *BAP1* LOF mutations:

However, this finding should be interpreted with caution as this comparison was made between 12 tumors with *BAP1* LOF mutations and 13 tumors without *BAP1* LOF mutations in the CheckMate cohort. There were insufficient numbers of patients to evaluate this question in

the TCGA cohort. As such, these findings do not alone call into question the established relationship between *BAP1* mutations and immune inflammation in RCC in general, but rather this only serves to reinforce that the immune-inflamed phenotype of S/R RCC tumors is not *solely* driven by *BAP1* mutations.

Q5 **Related to the study they cite showing a link between BAP1 loss and inflammation, it would be preferable to adapt the empirically defined TME signature to evaluate the contribution of the stroma rather than the use of sarcomatoid and non-sarcomatoid cancer cell lines.**

Response We appreciate the reviewer’s suggestion. The empiric tumor microenvironment (eTME) signatures are signatures that attempt to infer TME-specific genes in RCC using matched tumors, tumorgrafts, and normal tissue samples from the same patients with RCC, as previously described (PMID: 29884728). In order to adapt the eTME to the current study, we used two eTME signatures defined by the Wang et al. study: the 3x and 20x eTME signatures that include genes that are expressed at least 3 and 20 fold higher, respectively, in the immune/stroma component than in the tumor.

We computed single-sample gene set enrichment analysis (ssGSEA) scores for each of these signatures in the CheckMate and TCGA cohorts independently and compared the expression of these signatures between S/R and non-S/R RCC. The following two figure components were added to **revised Figure S6**, with corresponding edits to the **revised Results** section (page 11, lines 261-268).

Consistently with our other analyses, we found that both eTME signatures (3x and 20x) tended to be increased in S/R compared to non-S/R RCC. This finding again confirms the immune-inflamed phenotype of S/R RCC compared to non-S/R RCC. These findings were only found to be statistically significant in the TCGA cohort, but not the CheckMate cohort, likely owing to the larger sample size of the TCGA cohort (59 S/R and 830 non-S/R) compared to the CheckMate cohort (39 S/R and 247 non-S/R).

Q6	The added value of the drug studies is unclear since no differential response between sarcomatoid and non-sarcomatoid lines was found.
Response	We thank the reviewer for the opportunity to clarify this issue. We have included the drug studies in the supplementary material of our manuscript out of transparency. We believe that all data, even “negative” data, is important and should be reported. This is, of course, by no means a primary emphasis of our study but provides original data to the RCC community to avoid duplication of efforts.
Q6	Authors should clarify the data and expand the discussion about differences across the cohorts they analyze. For example, in Fig S2 there are differences across the cohorts, but this is not so obvious given the different units and small axes font sizes.
Response	We thank the reviewer for their suggestion. The methods and results sections currently contain the details of the specificities of each cohort analyzed. In order to reinforce the reviewer’s key point that there exists heterogeneity between the analyzed cohorts, the following statement was added to the revised Discussion section: “We additionally acknowledge that the data used in this study originated from cohorts that differed in the types of samples used (such as frozen tissue vs. formalin-fixed paraffin-embedded) and sequencing platform (panel vs. WES). However, despite this heterogeneity, the differences between S/R and non-S/R RCC tumors were found to be consistent across the different cohorts.” However, we emphasize that each cohort was analyzed separately (with ensuing meta-analyses of the results across cohorts where appropriate). Therefore, the consistency of the results of our analyses across the heterogeneous cohorts reinforces that our findings are robust to this heterogeneity.

Referee #3 (Remarks to the Author):

Summary	Van Allen, Choueiri and coworkers report the molecular characterization of sarcomatoid and rhabdoid RCC in a comprehensive manner by comprehensive DNA and RNA sequencing across an impressive number of tumors, some of which come from a clinical trial. While there have been past reports molecularly characterizing S/R RCC none have performed it to this degree and in such a large cohort and in such cutting edge manner. The authors find that specific alterations are enriched in S/R RCC. For example, CDKN2A deletion, and MYC gene signature upregulation. Moreover, tumors are enriched in an inflamed phenotype with high PD-L1 expression and increased clinical benefit from ICI. The latter point holds great clinical import, is based on multiple datasets, including data from the CheckMate
---------	---

	trial, and will likely be practice changing (despite being retrospective).
Response	We are very appreciative of this kind comment.
Q1	Pg 5, Line 102. When describing the non-S/R RCC cohorts it would be good to indicate the numbers of tumors in each of the outlined cohorts.
Response	We are thankful for this suggestion. As the reviewer suggested, we have added the breakdown of tumors in each cohort. This is now reflected in the Results section including:  - Page 5 lines 104-108: “This DNA-sequencing cohort included one clinical trial WES cohort (CheckMate cohort; 69 S/R and 342 non-S/R), a retrospective analysis of an institutional panel-based sequencing cohort (OncoPanel cohort; 79 S/R and 395 non-S/R), and a retrospective pathologic review and analysis of a publicly available cohort (TCGA cohort; 60 S/R and 828 non-S/R).” - Page 7 lines 168-171: “We compared RNA-seq data between S/R (total N= 98) and non-S/R RCC (total N= 1077) in the TCGA (publicly available; 59 S/R and 830 non-S/R) and CheckMate (39 S/R and 247 non-S/R) cohorts independently (Methods; Table S4) using Gene Set Enrichment Analysis (GSEA)²¹.”
Q2	Multiple genomic features of S/R RCC found by the authors have interestingly been explored in a prior GEM model of RCC [i.e. VHL loss, CDKN2A deletion, MYC overexpression] (PMID: 28593993). It is interesting that the authors of that paper did not see a S/R RCC phenotype in the mouse model. It seems worth a couple sentences (perhaps within or at the end of the middle paragraph on page 14) in the discussion mentioning this and the likelihood that other genetic or epigenetic events (i.e. Tp53 mutation) are required for a full S/R phenotype.
Response	We thank the reviewer for the interesting comment. We have added the following paragraph to the revised Discussion: “A previous study in genetically engineered mouse models had found that MYC activation with CDKN2A deletion and VHL deletion together produce kidney tumors that closely resemble human clear cell RCC⁴¹. While the authors of that study did not report histological patterns that resembled sarcomatoid and rhabdoid features, it is likely that these features are necessary but not sufficient to produce S/R features, and that other genomic and epigenomic features are needed to produce these aggressive tumors.”

REVIEWERS' COMMENTS

Reviewer #1 (Remarks to the Author):

I was satisfied with the responses.

The authors thank the reviewer for accepting the revisions.

Reviewer #3 (Remarks to the Author):

The authors have addressed my concerns and this is a great addition to the field of RCC cancer biology.

The authors thank the reviewer for their kind words.